# Impact of Climate, Stand Growth Parameters, and Management on Isotopic Composition of Tree Rings in Chestnut Coppices

**Francesco Marini [1], Giovanna Battipaglia [2,\*], Maria Chiara Manetti [3], Piermaria Corona [3] and Manuela Romagnoli [1]**

[1] Dipartimento per la Innovazione nei Sistemi Biologici, Università Degli Studi Della Tuscia, Agroalimentari e Forestali (DIBAF) Via San Camillo de Lellis snc, 01100 Viterbo, Italy; f.marini@unitus.it (F.M.); mroma@unitus.it (M.R.)

[2] Università Della Campania "L. Vanvitelli"—Dipartimento di Scienze Ambientali Biologiche e Farmaceutiche (DISTABIF), via Vivaldi 43, 81100 Caserta, Italy

[3] Consiglio per la ricerca in agricoltura e l'analisi dell'economia agraria, Centro di Ricerca Foreste e Legno (CREA-FL), 52100 Arezzo, Italy; mariachiara.manetti@crea.gov.it (M.C.M.); piermaria.corona@crea.gov.it (P.C.)

\* Correspondence: giovanna.battipaglia@unicampania.it; Tel.: +39-0823274647

**Abstract:** Research Highlights: Chestnut trees' (*Castanea sativa* Mill.) growth and their responses to climate are influenced by stand-characteristics and managements. This study highlighted that chestnut tree-ring growth is not particularly influenced by climate, while minimum temperature showed a positive relation with both intrinsic water-use efficiency (WUEi) and $\delta^{18}O$. Background and Objectives: The aim is to check the responses of chestnut trees to climate conditions and the role of stand structure and management. Materials and Methods: Stands with 12–14-year-old shoots were studied using dendrochronological and isotopic ($\delta^{18}O$ and $\delta^{13}C$) approaches. Correlations with climate parameters were investigated and principal component analysis was performed using site-characteristics and tree growth parameters as variables. Results: Correlations between tree-ring width (TRW), tree-ring $\delta^{18}O$, and $\delta^{13}C$-derived intrinsic water-use efficiency (WUEi) revealed stand-dependent effects. The highest Correlations were found between climate and tree-rings' isotopic composition. Chestnut was sensitive to high-minimum temperature in March and April, with a negative relationship with TRW and a positive relationship with WUEi. $\delta^{18}O$ signals were not significantly different among stands. Stand thinning had a positive effect on WUEi after 1–2 years. Stand competition (indicated by shoots/stump and stumps/ha) positively influenced both WUEi and $\delta^{18}O$.

**Keywords:** tree-ring width; climate change; water-use efficiency; $\delta^{13}C$; $\delta^{18}O$; chestnut tree physiology

## 1. Introduction

The European chestnut (*Castanea sativa* Mill.; Fagaceae) has a long tradition in Europe, the first written evidence of chestnut management is given in Theophrastus in the fourth century BC. Nowadays, the chestnut area devoted to timber production is 1.78 million hectares, corresponding to 66% of the total chestnut-growing area [1], mainly in Italy, Spain, Portugal, France, and Switzerland. Management systems and intensities highly vary among countries, although most of the area (79%) is managed as coppice stands [1]. Traditional chestnut coppices consist in short-rotation systems (5–25 years according to the targeted specific product) for the production of small to medium-sized poles [2]. This type of management allows to maximize the very high resprouting capacity of the stools and the remarkable initial growth-rate of the shoots, as well as recognizing the suitability of the chestnut tree

as a multipurpose species for landscape enhancing and conservation in rural, mountain, and marginal area [3].

In Italy, chestnut coppices (589, 362 ha) are important not only for wood production [4–7], but also because they represent an essential element of the culture and rural landscape, providing protection and ecosystem services essential for human well-being and conservation of resources [2,8].

The species shows remarkable plasticity [9], although adequate ecological conditions must be guaranteed for its healthy development. In particular, the annual minimum rainfall should not be less than 600 mm, with optimal growth conditions found in areas where precipitation is above 900–1000 mm. The species can withstand long periods of drought only if it grows in soils able to retain high moisture content, thereby reducing negative effects on tree growth [10]. The current trend in climate, with a considerable increase in water stress during summer across the entire Mediterranean Basin [11], could cause a loss of growth and wood productivity, in particular, in stands located at lower altitudes.

Silviculture and thinning practices can mitigate the effect of climatic stress. Thinning can have a positive effect on tree growth, improving stand resilience and resource use efficiency [10,12], as well as the physiological responses of trees to drought [13]. Thinning in chestnut has also been proven to increase wood quality conditions [4,5].

Tree-ring features, such as tree-ring width (TRW) and tree-ring carbon and oxygen stable isotope composition, have proved to be powerful tools when investigating the impact of climate on the growth and physiology of tree species. Indeed, stable carbon isotopes ($\delta^{13}$C) is mainly controlled by stomatal conductance during carbon fixation (gs) and the rate of photosynthesis (A), both of which are driven by environmental conditions [14,15]. $\delta^{13}$C in tree rings has been used to study long-term aspects of tree physiology in addition to sensitivity to climatic parameters. For example, an increase in $\delta^{13}$C could indicate stomatal closure and reduced conductance to prevent water loss during drought [16,17] or could be due to changes in photosynthetic rates affected by irradiance during cool and wet periods' coppices [18]. This research aims to verify the responses of chestnut trees to climate conditions and the role of stand structure and management. Five stands, differing for shoots and stumps number and density, have been selected on Monte Amiata. We hypothesized that tree growth and intrinsic water-use efficiency (WUEi) would be influenced by stand characteristics and by the applied management, with thinning practice, that could have a positive effect on productivity.

## 2. Materials and Methods

### 2.1. Site Description

The study was conducted on Monte Amiata, a lava dome located in the Province of Siena, Tuscany, central Italy, with an altitude between 990 and 1145 m a.s.l. (Figure 1). The site is characterised by medium to high fertility due to the volcanic matrix of the soils. The parent material is a trachyte lava with a high silicate content and poor in basis, the slope is generally gentle, but a few outcrops are present, and erosion and landslides are absent. Brown and sub-acid soils of good physical structure are prevalent. According to the map of the Tuscany soil (http://sit.lamma.rete.toscana.it/websuoli/), the soil is *Andic Dystrudepts* coarse-loamy, siliceous, mesic, deep, very soft, non-gravelly, and well drained.

Monte Amiata is an important area for chestnut cultivation because of the high growing stock and the economic relevance of wood and fruit production for local communities. Chestnut forests cover 7500 ha in Monte Amiata, with half the area (3500 ha) managed for wood production under public ownership. The forests are located mainly in the eastern side of the region at 800 to 1200 m a.s.l. Chestnut forests tend to be monospecific and are of anthropic origin, and secondary species are rare or absent. In Italy, chestnut forests are usually managed by coppicing in a coppice-with-standards system, with the rotation age varying between regions. Forest management is different depending on ownership. In forests under private ownership, rotation is 16–20 years without thinning, with the number of standards greater than 100/ha. In forests under public ownership, rotation is usually 24–30 years, and there are generally two thinning events, the first when the coppice is 6–8 years old

and the second when it is 12–15 years old. The number of standards is usually 30–50/ha. For economic reasons, the first thinning is usually not carried out. The stands investigated in this study are under public ownership.

## 2.2. Climate

Climate data for the period 2004–2016, recorded at the Le Vigne meteorological station in Piancastagnaio, Province of Siena (450 m a.s.l.; 723370 E UTM, 4744226 N UTM), about 2.5 to 4.5 km from the investigated stands, show an annual rainfall of 1117 mm and an annual average temperature of 13.2 °C.

The Bagnouls–Gaussen diagram (Figure 1) for the period 2004–2016 shows a positive peak in precipitation in November as well as an absence of drought during the summer. Maximum temperatures were reached in July and August, and minimum temperatures in January and December. Precipitation decreased in April, followed by an increase in May. The sum of precipitation data shows an oscillating trend during the 2004–2016 period (Figure 2), with two significant negative peaks, in 2007 (one of the most severe droughts in recent years [19]) and in 2011. Positive peaks occurred in 2008, 2010, and 2014. Mean annual temperatures show an increasing trend, especially in the last few years.

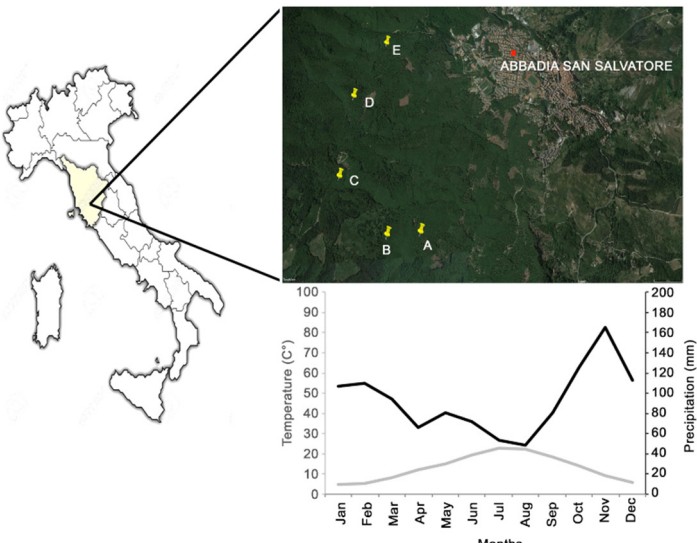

**Figure 1.** Study site in central Italy and Bagnouls–Gaussen climatic diagram of Piancastagnaio (SI) meteo station related to the period 2004–2016.

We studied five different stands of young chestnut coppices 12–15 years in age. The stands were Sant'Antonio (stands A and B), Cipriana (stand C), Le Decine (stand D) and Acquagialla (stand E).

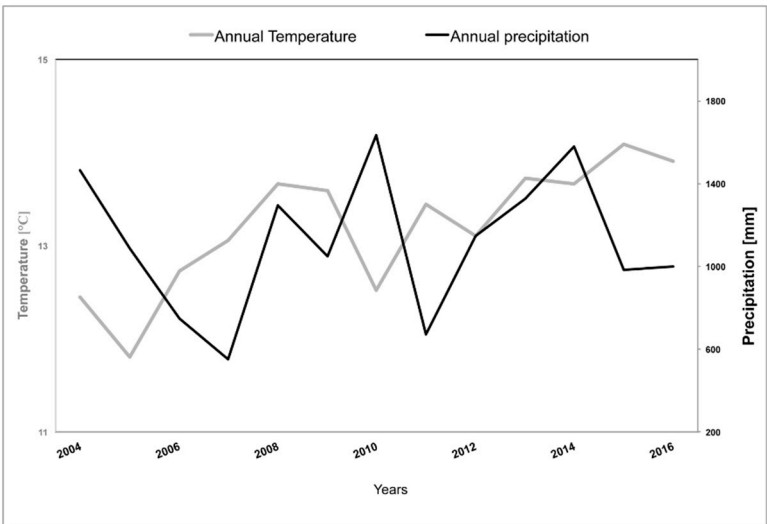

**Figure 2.** Climatic diagram of the mean annual temperature and total amount of precipitation for the Piancastagnaio meteo station.

## 2.3. Growth Stand Conditions

The main dendrometric parameters were measured on five representative circular sampling plots with a 15 m radius. The parameters were number of stumps, number of stems, diameter at breast height (DBH), mean height of stems, and the number of standards in the coppice, with their diameters and heights. In each stand, 15 old dominant shoots from different stumps were randomly selected and a disc from each stem was sampled at 1.30 m above ground.

## 2.4. Dendrochronological Analysis

Tree-ring width (TRW) was measured using a LINTAB device (RINNTECH, Heidelberg, Germany), to an accuracy of 0.01 mm, using a Leica MS5 stereomicroscope. Two orthogonal radii per disc/shoot were measured. The 15 tree-ring series from each stand were graphically and statistically cross-dated according to standard procedures in dendrochronology [20]. Because the series were short, correlation coefficients and statistical synchronization between individual series were considered. A mean curve was built using TSAP-WIN Professional software (RINNTECH) representative of each stand, averaging the raw ring-width measurements. Descriptive dendrochronological parameters were calculated, i.e., mean TRW (and standard deviation) and mean sensitivity, a measure of the mean relative change in the width of adjacent rings, calculated as the ratio of the difference in ring width between year t + 1 and year t, and their average according to the following formula:

$$\frac{(\mathrm{TRWt} + 1 - \mathrm{TRWt})}{\left(\frac{\mathrm{TRWt+1} + \mathrm{TRWt}}{2}\right)} \tag{1}$$

The autoregressive moving average (ARMA) model [21] was used to remove growth trends and enhance the climate signal in the raw data for TRW. The model is frequently used in dendroclimatological analysis [19,22], and is based on the estimation of the autocorrelation functions describing year-to-year changes in annual ring width. The term AR refers to the order of the autoregressive model applied to the tree ring series, while MA refers to the order of the moving average model. The autocorrelation analysis of the tree-ring series suggested the selection of model 2.0, considering simple correlation and the partial autocorrelation profile, in the software PAST (PAleontological STatistics, University of Oslo, Oslo, Norway). Indeed after the second order, the autocorrelation between tree rings becomes negligible, whilst the moving average model in our investigation was not applicable due to the

shortness of the tree ring series. Mathematically, AR2 and MA0 in the ARMA model can be expressed as:

$$TRW = \varphi_1 TRW_{t-1} + \varphi_2 TRW_{t-2} + a \tag{2}$$

where a is the residual part obtained by subtracting the modelized part ($\varphi_1 TRW_{t-1} + \varphi_2 TRW_{t-2}$) from the TRW raw data. It corresponds to the random variation which is related to the action of climate. The coefficient $\varphi$ is the autocorrelation value and it ranges from 0 to 1.

The dendroclimatological analysis was carried out using climate data from the Le Vigne meteorological station, i.e., monthly precipitation (mm) and average monthly minimum and maximum temperatures (°C). The data were organized from October of the previous year (t − 1) to September of the current year (t) into a dataset of 12 monthly independent variables, obtaining three different matrices: monthly precipitation, monthly maximum temperature, and monthly minimum temperature.

*2.5. Dendroisotopic Analysis*

In each of the five stands, five discs were selected for isotope analysis using the Gleichläufigkeit (GLK) measure, which evaluates the proportion of agreement/disagreement in interannual growth tendencies between trees [23]. The five selected discs showed the best cross-dating (GLK > 70%) with the corresponding average plot chronology. The annual growth rings formed in 2004–2016 were manually separated from each other with a razor blade under a microscope. The annual wood ring was milled and homogenized in a hammer mill (MF 10 basic Microfine grinder drive, IKA-Werke, Staufen im Breisgau, Germany), using a mesh size of 0.5 mm. Cellulose extraction was avoided since it has been demonstrated to not be necessary for ecophysiological and dendrochronological studies that analyze the response of trees to environmental changes recorded within the sapwood, i.e., in a relatively short period [24–26]. The $\delta^{13}C$ and $\delta^{18}O$ compositions of tree rings were measured at the IRMS Laboratory of the University of Campania "Luigi Vanvitelli", following standard procedures [27].

The analysis of stable carbon isotopes in tree rings ($\delta^{13}C_{TR}$) is a powerful tool for estimating the magnitude of carbon Discrimination ($\Delta^{13}C$):

$$\Delta^{13}C (‰) = (\delta^{13}C_a - \delta^{13}C_{TR})//1 + \delta^{13}C_{TR}/1000) \tag{3}$$

where $\delta^{13}C_a$ is the isotopic value of atmospheric $CO_2$ and can be estimated for the period 1990–2003 from Reference [28], and measured values for the period 2004–2016 are available online (http://www.esrl.noaa.gov/gmd/).

Following Farquhar et al., carbon discrimination during $CO_2$ fixation of C3 plants is linearly related to the ratio of intercellular to atmospheric ($CO_2$) ($c_i/c_a$) by the equation:

$$\Delta^{13}C (‰) = a + (b - a) c_i/c_a \tag{4}$$

where a is the fractionation for $^{13}CO_2$ as a result of diffusion through the air (4.4‰) and b is the fractionation during carboxylation (27‰). Combining Equations (3) and (4) $c_i$ can be calculated using $c_a$, the concentration of $CO_2$ in the atmosphere, estimated for each year and obtained by the National Oceanic and Atmospheric Administration- NOAA (Mauna Loa station; http://www.esrl.noaa.gov/).

Intrinsic water use efficiency ($WUE_i$), is defined as the ratio of assimilation rate (A) to stomatal conductance for water vapor ($g_w$) [29]:

$$WUE_i = A/g_w \tag{5}$$

Since $g_w$ = 1.6 $g_c$ ($g_c$ is the conductance for $CO_2$), and given that the net carbon uptake by diffusion through the stomata (A) follows Fick's law,

$$A = g_c (c_a - c_i) \tag{6}$$

Then, we calculate WUE$_i$ combining Equations (4)–(6)

$$WUEi = A/(gc\ 1.6) = (ca - ci)/1.6 = ca(b - \Delta^{13}C)/1.6(b - a) \qquad (7)$$

Several studies suggested caution in simplistic interpretations of WUEi based on interspecific variation in $\delta$ $^{13}$C, when comparing plant species with wide-ranging leaf anatomy or physiology [30,31]. However, it has been demonstrated that $\delta$ $^{13}$C can be used as a 'relative index' of WUEi in order to rank species occupying the same environment [27,32–34].

### 2.6. Statistical Analysis

Data were analyzed using PAST 3 software. Since the data were not normally distributed, dendrometric differences between the five studied stands were explored through a post-hoc nonparametric test (Mann–Whitney U test) of the dendrometric variables: number of shoots in each stump (SS), shoot dominant height, number of stumps per hectare, and stand volume per hectare (volume/ha).

Moreover, principal component analysis (PCA) was used to obtain information on dendrometric variables across the test sites.

The matrix was built considering two individual parameters for each shoot, basal area (g) and the number of shoots in the sample's stump. To the shoots of the same stand, the dominant height, the number of shoots/ha, and the number of stumps/ha were associated.

Finally, Pearson's correlation analysis was performed to assess the relationship between the independent variables, monthly climate data from October of year t − 1 to September of year t, and the dependent variables, mean isotopic series of the five stands and the residual AR model for TRW over the 2004–2016 period. We performed correlations with $\delta^3$C corrected data to verify the coherence with WUEi correlations, and since the main results did not change, we decide to show only the latest.

## 3. Results

### 3.1. Dendrometric Characteristics

The main dendrometric parameters of the five chestnut coppice stands on Monte Amiata are reported in Table 1.

**Table 1.** Dendrometric characteristics of the studied stands of chestnut at Monte Amiata. Sd = standard deviation, dbh = diameter breast height, G = basal area.

| Stand | A | B | C | D | E |
|---|---|---|---|---|---|
| Altitude (m) | 990 | 1030 | 1145 | 1100 | 1030.0 |
| Exposure | SE | SE | SE | SE | SO |
| Slope (%) | 0 | 30 | 0 | 30 | 40.0 |
| Age (years) | 15 | 11 | 12 | 15 | 11.0 |
| Shoots' Dominant height (m) | 14.9 | 15.3 | 13.6 | 13.8 | 14.0 |
| Stumps ha$^{-1}$ | 580 | 623 | 780 | 1047 | 962.0 |
| Shoots ha$^{-1}$ | 3480 | 3860 | 4810 | 2845 | 5210.0 |
| Shoots/stumps | 9.8 | 8.25 | 8 | 4.2 | 9.1 |
| *sd* | 3.6 | 2.4 | 4.5 | 1.5 | 4.0 |
| Shoots dbh (m) | 9.6 | 10.6 | 8.9 | 10 | 7.0 |
| *sd* | 2.9 | 2.91 | 2.87 | 2.02 | 2.0 |
| Stumps/ha | 70 | 113 | 56 | 70 | 110.0 |
| Volume (m$^3$ ha$^{-1}$) | 193 | 269 | 220 | 156 | 136.0 |
| G (m$^2$ ha$^{-1}$) | 25.41 | 34.20 | 29.95 | 22.42 | 20.31 |
| Tree Ring Width (1/100 mm) | 372.95 | 355.34 | 403.78 | 314.75 | 338.2 |
| *sd* | 147.75 | 172.74 | 140.75 | 98.34 | 125.8 |
| Mean Sensitivity | 0.36 | 0.3 | 0.27 | 0.31 | 0.4 |

To sum up, the most important outcomes were that stand D, thinned in 2014, was characterized by the lowest stem number (2.7 stems/stump), low basal area, and number of stumps/ha. Despite the apparent high growth potential, as assessed from the dominant height of the stems (14.0 m at 11 years), stand E exhibited low productivity in terms of basal area (20.31 m2 ha⁻1) and reduced growth (mean Diameter Breast Height = 7.0 cm).

The nonparametric Mann–Whitney U test confirmed that the number of shoots/stumps (SS) in stand D was significantly lower than the SS of the other stands examined (Table 2). Furthermore, stand E had a significantly lower mean DBH and shoot basal area. The widest tree ring was in stand C, the narrowest in D, and the highest mean sensitivity was in A and E (see Table 1). There were no significant differences in dendrometric variables between A, B, and C stands.

The PCA highlighted the results obtained by the nonparametric test (Figure 3), with stand C located at an intermediate position with respect to the other stands in terms of dendrometric variables, showing the widest distribution of shoots along the PCA axes.

**Table 2.** Mann–Whitney analysis of different dendrometric characteristics of the five stands.

|   | **A** | **B** | **C** | **D** | **E** |
|---|---|---|---|---|---|
| A |   |   |   | Di * g * SS *** | Di *** g *** |
| B |   |   |   | SS *** | Di *** g *** |
| C |   |   |   | SS | Di *** g ** |
| D |   |   |   |   | Di *** g ** SS *** |

*** = $p < 0.001$; ** = $p < 0.01$; * = $p < 0.05$. Di = diameter; g = shoot basal area; SS = shoots/stumpss.

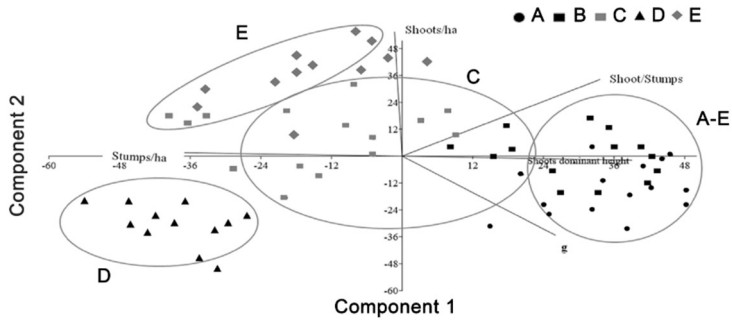

**Figure 3.** Principal component analysis (PCA) of dendrometric variables.

E and D stands were distinct from the other stands. A and B stands were in the same group, and were quite close, because they were very close to each other. With the first two main components, the PCA was able to explain 67.72% of the variance. The first principal component was associated positively with shoot dominant height and negatively with stumps/ha, whereas the second principal component was associated with the variable shoots/ha. The number of shoots/stumpss determined the different position along the two PCA axes for stand E.

### 3.2. Dendrochronology and Dendroclimatology

The TRW series revealed a similar trend among all stands, with a strong decrease in TRWs, which is characteristic of coppice stands where regeneration is mainly vegetative, and growth is highly related to boosting by the stump (Figure 4a). Interannual variability was quite low. The mean curve for stand D was below the curves for the other stands, and the TRWs of the stand were comparable to those of the other stands in 2011 (the dendrochronological curve for stand D crossed that for stand B, and it reached TRW values of the A, B, and E stands). A weak effect of the thinning carried out in 2014 was

seen as a subsequent increase in ring width. Positive TRW peaks common to all stands occurred in 2009 and 2015, while common negative peaks appeared in 2010, 2014, and 2016.

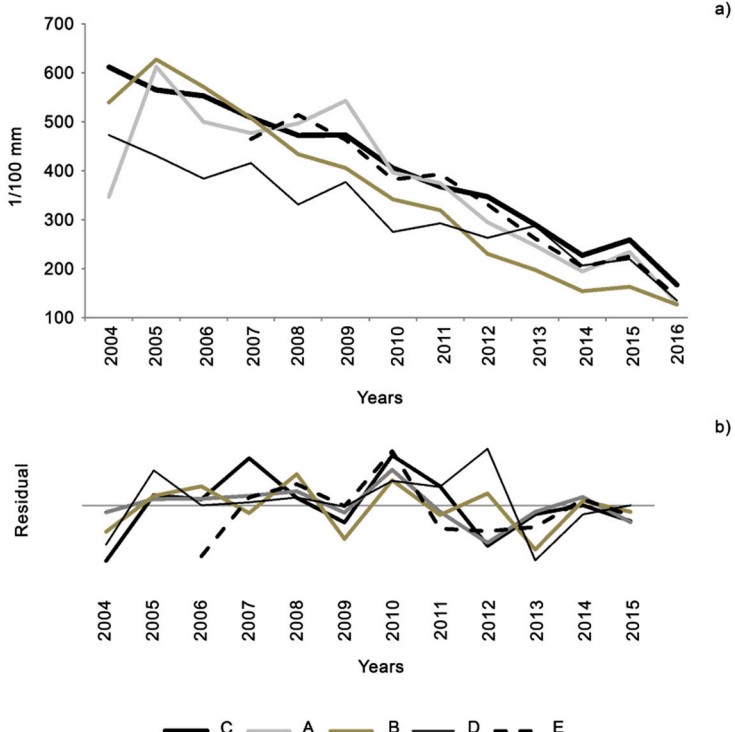

**Figure 4.** (**a**) Raw tree-ring width (TRW) series of the five stands of chestnut coppices at Monte Amiata, (**b**) Residual values of TRW by the ARMA model.

The correlation function of the AR residuals of TRW and climate parameters in all five stands (Figure 5) shows that chestnut tree-ring growth is not particularly influenced by climate (the correlation coefficients were not significant). Growth was affected mostly by minimum temperature (Figure 5a) rather than by maximum or mean temperature (data not shown). Only stands A and B showed significant correlations between climate and growth. TRW in A had a significant positive correlation with December (t − 1) precipitation (r = 0.59), and stand B had a significant positive correlation with minimum temperature in November (t − 1). Interestingly, both stands were characterized by the lowest number of stumps/ha (see Table 1). In stand E, a reduction in tree ring growth was associated with high precipitation in April, while increased growth may be ascribed to the high mean temperature in May. However, even if the relationship was not significant, a negative correlation with minimum temperature in March and April and a weak correlation with precipitation in December of year t − 1 were reported in all stands.

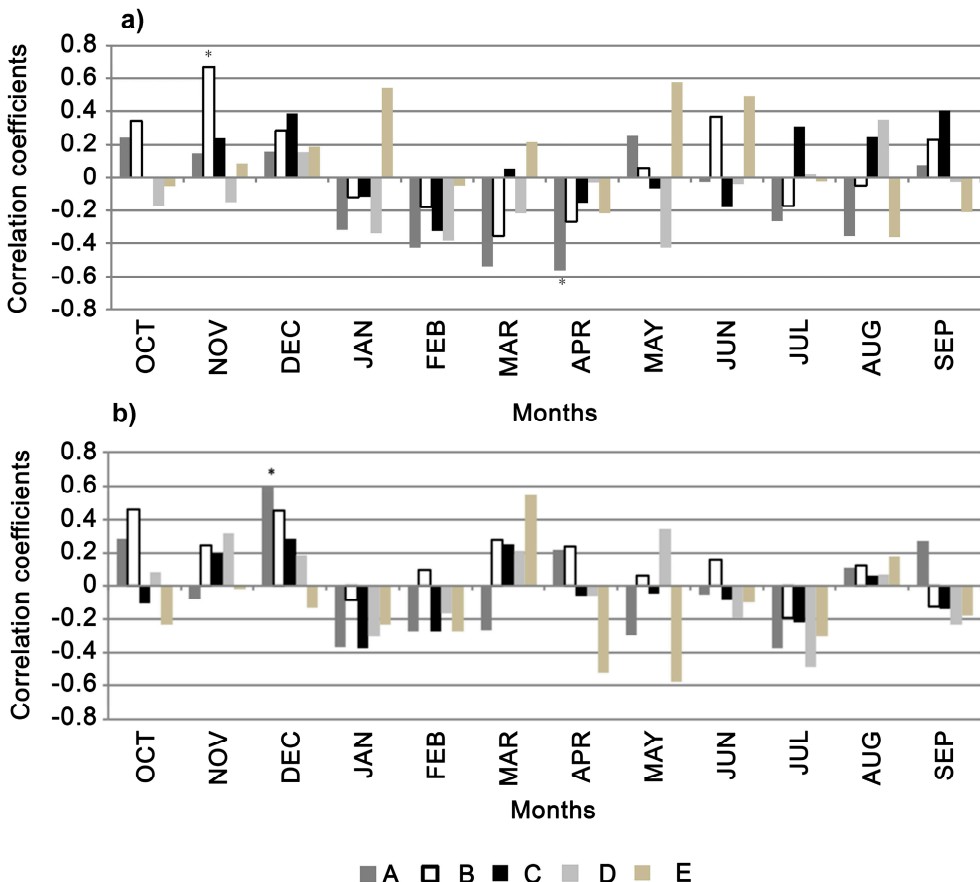

**Figure 5.** Correlation function of TRW with (**a**) mean minimum temperature and (**b**) precipitation for each stand.

### 3.3. Stable Isotopes

Figure 6a shows the temporal variation in $\delta^{18}O$, with a common minimum or negative annual trend during 2012 and 2014. Figure 6b shows the WUEi values, calculated from $\delta^{13}C$. The curve indicates a generally increasing trend in the last growth years, with a peak in 2015. At stand level, the E WUEi curve lay below the other stand-specific mean curves until 2013–2016. WUEi for stands A and D (Figure 6b) increased, and D was thinned at 2014, showing an opposite trend in comparison to the curves for the other stands.

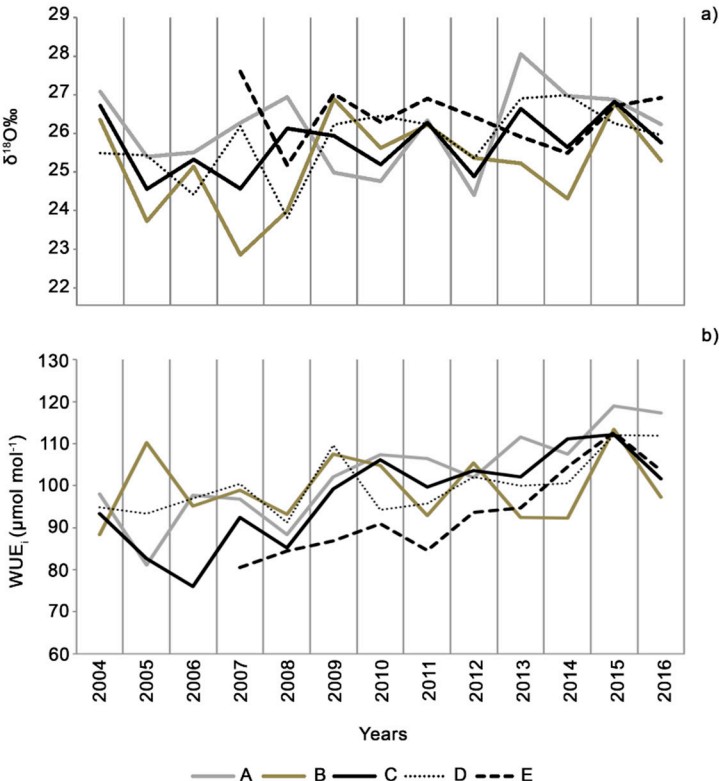

**Figure 6.** (**a**) $\delta^{18}O$ time series and (**b**) intrinsic water-use efficiency (WUEi) time series for each stand.

$\delta^{18}O$ values of the five stands (Figure 6a) showed high annual variability during the study period, with a positive peak in all the stands in 2015 except in the stand D. WUEi shows an evident increasing trend in the stands A, C, and E, and in the stand D, an abrupt increase is in 2015, which is prolonged in 2016. Analysis of variance (ANOVA) and correlation tests were performed to assess the differences between stands in all stable isotope data (data not shown). Only one significant result was obtained—$\delta^{18}O$ differed between stands E and B. This means that, in general, neither WUE$_i$ nor $\delta^{18}O$ differed significantly between stands.

The most representative correlations between climate parameters and stable isotopes in chestnut (Figure 7) were those between WUEi and minimum temperature, which showed the most frequent and significant responses. The response to minimum temperature was generally positive for both WUEi and $\delta^{18}O$. The most important signals were the significant positive correlations between WUEi and October–November (t − 1), March–April, and June–July minimum temperatures. Based on the $\delta^{18}O$ data (Figure 7d), the temperature effect appears to have been delayed, because there was also a positive correlation in August and September (Figure 7b). Less agreement was observed in the response to precipitation between the different stands for both isotopic values.

At the stand level, we observed a strong positive correlation between WUEi and December (t − 1) minimum temperature for stand B, a positive correlation between $\delta^{18}O$ and March precipitation for C and A, a positive correlation between $\delta^{18}O$ and February precipitation for D, and a strong negative correlation between $\delta^{18}O$ and September precipitation for D and E.

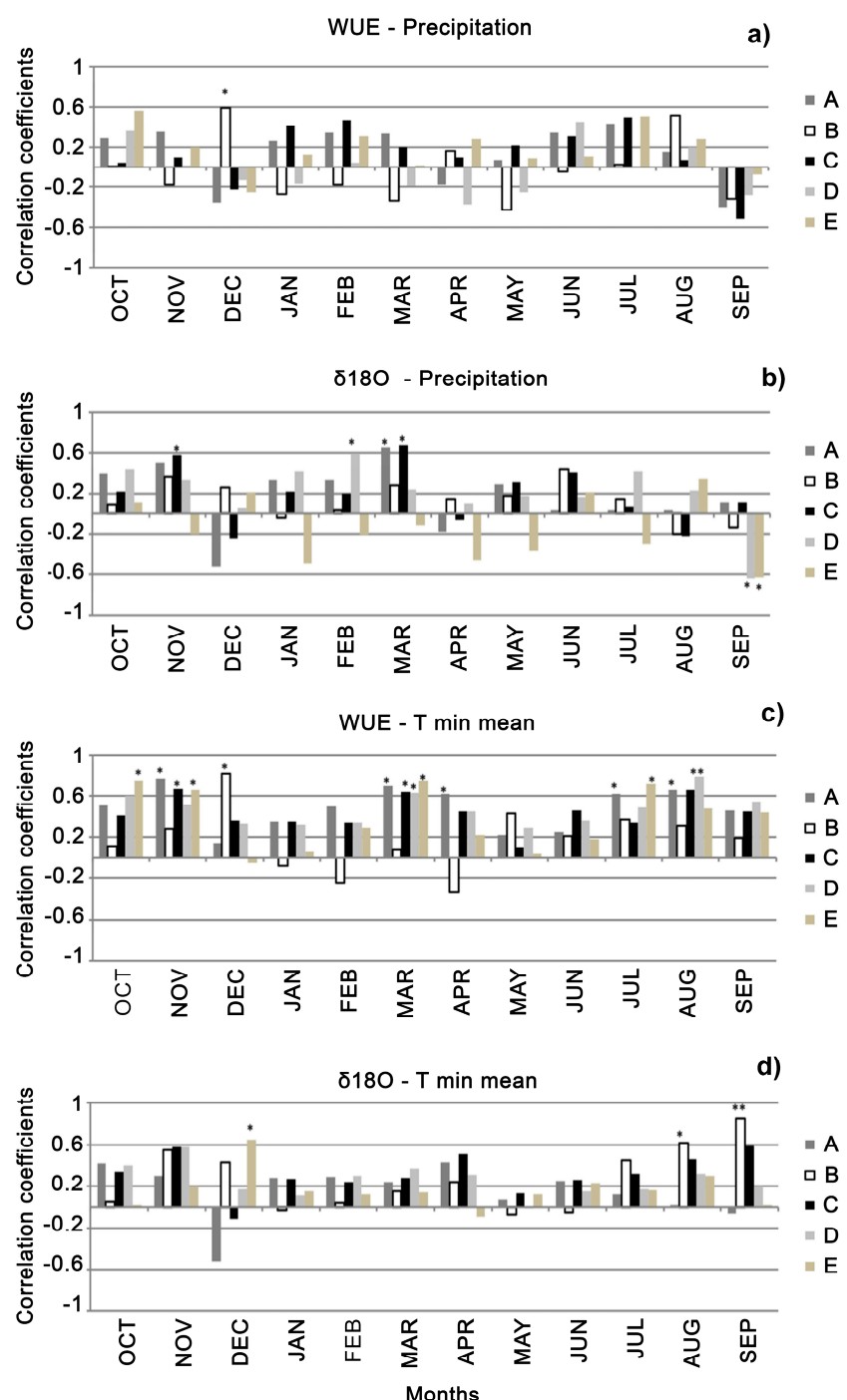

**Figure 7.** Correlation among climate parameters and stable isotopes.

## 4. Discussion

Species signals:

The dendrochronological series of the examined stands were short, and a weaker climate signal was expected due to the strong impact of vegetative regrowth from stumps. Chestnut is characterized by relatively wide tree rings [4–6], and it is not a particularly sensitive species in terms of dendrochronology [35]. Moreover, is not studied much from the dendrochronological point of view [36], and physiological studies of the species are also quite rare [9,37,38]. Based on dendrometric parameters, the selected stands in our study were found to be quite different from each other, mainly in the number of stumps/ha and the number of stems/stumps. Different dendrometric parameters affect

chestnut stem competition, and they may also influence the effect of climate parameters on growth. The PCA analysis in this study highlighted a different situation in stand E (highest number of stems, lowest DBH, and volume/ha) compared to stand B (highest DBH, higher number of stems/stumps, greatest dominant stem height, few stumps/ha) and D (Figure 3).

Tree rings are a valuable archive of environmental information because wood formation is regulated not only by intrinsic factors, such as gene expression and hormonal signals, but also by climate variables, such as temperature and precipitation [39–45].

In this study, TRW was not particularly sensitive to climate. In contrast, tree-ring isotope composition showed a stronger relationship with climate parameters. Previous studies showed that isotopes seem to be more sensitive to climate because they allow more efficient capture of signals of the plant–environment interactions [46]. There are no previous records of $\delta^{18}O$ and $\delta^{13}C$ for chestnut, in addition, very few studies have dealt with chestnut wood characteristics in relation to climate parameters [9,37,47–49]. Thus, by analyzing the relationships between climate parameters and TRW and stable isotopes, it was possible to extract common signals at the investigated stands, which may be related to the ecology of chestnut. Minimum temperatures in March and April were positively related to WUEi in almost all the stands (except stand B) and $\delta^{18}O$, while a negative correlation resulted between minimum temperatures and TRW in the same period. This latter result agrees with the studies of Genova and Gracia [48], which also showed a negative correlation between March temperature and TRW. As reported in Cufar et al. and Romagnoli et al. [9], the first earlywood vessels develop before the resumption of photosynthetic activity, and their formation depends on site characteristics and climatic conditions of the current year. At the latitudes of central Italy, earlywood formation starts in the first two weeks of April, and the onset of cambial division is estimated to be in the first days of April [9]. Warm minimum temperatures hasten the onset of growth and prevent freezing-induced embolism in oak species with ring-porous wood [50–53], thereby boosting the formation of earlywood vessels, which are mainly responsible for resource use efficiency in ring-porous species [52]. Fonti et al. [37] also showed a positive correlation between the formation of earlywood vessels and minimum temperature, and determined initial vessel appearance. Castellani [54] estimated that the radial growth of Italian chestnut at the latitudes of the stands we examined started from March when the mean temperature exceeded 8 °C. If the minimum temperature is too high in Monte Amiata, earlywood vessel size is reduced, causing a decrease in TRW.

During the full growing season in late spring and midsummer, chestnut is not affected by the amount of precipitation, which is not particularly limiting in the Monte Amiata area. In July (stands A and E) and August (stands A, C, and D), WUEi shows a positive correlation with minimum temperatures. Latewood formation and lignification in chestnut occur in that period [9,55], and it is well known that cell wall thickening and lignification rates increase with temperature [53]. Although water is not thought to be a limiting factor in this area, periods of drought stress with a risk of cavitation can result in a small number of latewood vessels [45] and more fiber cells where carbon is stored. Furthermore, the increase in WUEi depends on stomatal sensitivity which seems, in general, greater in ring-porous than in diffuse-porous species [56].

Mean temperatures and precipitation in autumn positively affected TRW in the following year in stands A and B, respectively. The effects of previous autumn–winter climatic conditions on the growth of ring-porous species have been evaluated in oak [21,22,45]. Carbon allocation to tissues and water accumulation in soils increase earlywood production with the carbon stored during the previous growing season [57]. Interestingly, in previous studies, a negative correlation between WUEi and winter temperature has been shown for ring-porous species; however, this contrasting pattern is due to the drier conditions prevailing at the analyzed site, causing high temperatures to induce high-respiration rates as well as consumption of stored carbohydrates [57].

A global increase in WUEi in the last few decades has been recorded, based on WUEi curves. The phenomenon has been ascribed to the global $CO_2$ increase [58]. Although the trees we studied were generally in a healthy condition, the increase in WUEi may indicate a common adaptation to

drought in Mediterranean species [16,59]. Since our trees are very young and they all experienced the same environmental conditions growing in the same site, Monte Amiata, we prefer to focus on the comparison between the stands, instead of speculating too much on the absolute value of WUEi increase.

The $\delta^{18}$O data are more stable through time, and the correlation between $\delta^{18}$O and climate parameters is less significant. The oxygen isotope composition of tree rings is largely determined by the isotope composition of source water (i.e., precipitation) and evaporative effects on leaf water enrichment [28,60,61]. The stability of the $\delta^{18}$O signal may mean that the plant has access to the ground-water table with little variability in $\delta^{18}$O [62]. In addition to high transport efficiency due to large vessels, the main traits of ring-porous species are lower responsiveness of transpiration to soil drying and higher potential for water uptake from underground sources due to a deeper root system. The behavior of chestnut is quite similar to that of other ring-porous species because of its isohydric response and the deep root system of stumps [52], rendering the species less sensitive to possible climatic stress. However, the source water signal can be modified by large variability in evaporative enrichment in drought-adapted Mediterranean species with tight stomatal regulation of transpiration [59], and stomatal regulation may be seen as one possible functional adaptation against drought. Other parameters can affect climate–isotopic content relationships. Reynolds-Henne et al. [55] ascribed the variance in tree-ring $\delta^{18}$O not to a simplistic effect of temperature, but to its interaction with the ecological setting and physiological and biochemical properties of the tree.

Stand signals:

At the stand level, E had the highest number of stems/ha and a high number of stumps/ha, so that it had the worst conditions for resource competition. Stand E was significantly climate-dependent compared to the other sites, showing significant correlations between WUEi and the climate parameters (Figure 7a). The mean WUEi curve for stand E was below the curves for the other stands (Figure 6b), while its TRW was comparable (see Figure 4a). Therefore, despite the apparent high competition, tree-ring growth, as assessed by TRW, was not compromised, probably because of water availability. However, there was an abrupt change in the slope of the WUEi curve for E in the last three years, possibly related to the increase in minimum temperature in this period, making it necessary for trees in this site to adopt a strategy to increase their water-use efficiency. The PCA (Figure 3) showed that the dendrometric parameters of A and B stands were very similar. They were the most productive stands, but the correlations between their isotopic content and climate showed some dissimilarities.

Stand B was characterized by a small number of stumps/ha and the highest number of stems/stumps. In this site, WUEi was highly related to both minimum temperature and precipitation in December t − 1. The high minimum temperature of autumn t − 1 may have facilitated carbohydrate allocation to tissues [63], which may in turn have supported the formation of a wider tree ring in the following spring. The smaller number of stumps in this stand may have allowed for more efficient water accumulation during winter (positive correlation with precipitation), due to less competition between the root systems of the stumps, compared to the other stands. The smallest number of stumps occupying the stand, together with a high number of shoots/stumps, may have made B the only stand sensitive to temperature, as determined by the positive correlation between δ18O and minimum temperature. In fact, $\delta^{18}$O showed a positive correlation with minimum temperature at the end of the summer when chestnut wood formation had almost ended [9] but the lignification process was still going on in some stands. The higher $\delta^{18}$O may indicate a state of water stress, which may have been caused in part by a low canopy layer and by some site-specific conditions, such as an evident slope [56], which was not present in stand A.

Stands A and C are comparable in their responses to climate. They showed a species-specific positive response of WUEi to minimum temperature, but they were also characterized by a positive correlation between $\delta^{18}$O and March precipitation. Higher precipitation in the study site may be related to late-winter frost damage which causes early cavitation [64].

Stand D was located in the only site where thinning was carried out in 2014, two years before sampling. Currently, the stand has the lowest number of stems/stumps. After 2014, D (Figure 6b) showed an increase in WUEi that may be attributable, partially, to a higher overall increase in photosynthetic efficiency [31,33], as seen in ring-porous species. Indeed, even if the increase in WUEi was also recorded in other stands, such as A, stand D starts to increase WUEi as soon as after the thinning practice and not before. Further, this increase has not resulted up to now in an evident increase in productivity, as has been demonstrated for other Mediterranean species [58]. Stand D generally did not show any correlations between TRW and climate; thus, the stand it is located in appears not to be significantly sensitive to climate. However, the WUEi of the stand was highly susceptible to the high minimum temperature in August, a period characterized by the driest conditions. Furthermore, in the thinned stand D, a slight increase in ring width, which reached the values of the other stands, was observed during 2015, testifying to the effect of thinning. Information on the role of thinning in WUEi is scarce and contradictory [65]. For example, di Matteo et al. [38] showed an increase in $\delta^{13}$C after thinning in *Quercus cerris* L., another ring-porous species, while Martin-Benito et al. [66] reported no variation in WUEi after moderate thinning. Therefore, whether thinning has an impact on isotopic content and water use efficiency needs further exploration and a long-term study.

## 5. Conclusions

This is the first study to focus on the dendroisotopic composition of chestnut trees, and the results contribute to a better understanding of the physiology of the species. Stable isotopes in chestnut proved to be much more sensitive to climatic variation than tree-ring width. They provided better information on stress factors and possible indications of best practice for forest management to avoid the loss of physiological performance and functionality. The selected coppice stands were quite distinct from each other, mainly in terms of the number of stumps/ha and the number of stems/stumps. Only one stand had been thinned, but common climate signals, which are related to characteristics of the species, were detected in the investigation. Climate was not particularly limiting for growth in the Monte Amiata area, although an increasing trend in minimum temperature was observed. All the stands were sensitive to minimum temperature, especially in March and April, with a positive effect on WUEi. The signal was related to the formation of earlywood vessels, which are the main transport elements in the species. During summer (July and August), the stands showed an increase in WUEi.

Past forest management practices are reflected in part in the relationships between isotopic content and climate. The stand with the highest competition between shoots and between stumps was the most sensitive to climate parameters, testified by an increase in WUEi in the latter years of the study. This stand would have needed to store carbohydrates in the previous autumn for growth; however, if temperatures in December were too high, carbohydrate consumption could have exceeded carbohydrate accumulation.

The only stand experiencing high transpiration at the end of the summer had low vegetation coverage on soil due to the small number of stumps/ha and high competition due to the high number of stems/stumps, as well as a significant slope absent in other stands.

Because of increasing minimum temperature over the last few years, it is expected that trees will need to adopt a strategy to decrease processes such as cavitation and embolism. The impact is expected to be more evident in stands characterized by high competition.

Thinning could play an important role, even at sites currently not experiencing strong climatic stress. Indeed, even if our dataset is not exhaustive, we could observe that thinning increases WUEi mainly during the summer months. The reason may be a more efficient use of water resources compared to stands with a denser canopy and a higher number of stems. This confirms the role of thinning for increasing water use efficiency in chestnut coppice stands under changing climatic conditions.



**Author Contributions:** Conceptualization all the authors. Methodology: F.M. and M.C.M. field measurement, G.B. isotopic analysis, F.M., M.R., statistical processing. Writing and editing F.M., G.B., P.C., M.R., M.C.M. All people listed as authors approved the final submitted version and agree to share collective responsibility and accountability for the work.

**Funding:** Funded by "Departments of Excellence-2018" Program (Dipartimenti di Eccellenza) of the Italian Ministry of Education, University and Research, DIBAF-Department of University of Tuscia, Project "Landscape 4.0—food, wellbeing and environment". The work has been also partially founded by CREA with a financial support to the PhD program of DIBAF.

**Acknowledgments:** Many thanks to the "Union of municipalities of Amiata and Val D'Orcia" and Pier Giuseppe Montini.

**Conflicts of Interest:** The authors declare no conflict of interest.

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
