# Peer review of "Impact of Climate, Stand Growth Parameters, and Management on Isotopic Composition of Tree Rings in Chestnut Coppices"

_forests, doi:10.3390/f10121148_

Round 1

Reviewer 1 Report

The article is mainly concentrating on dendroisotopic content of chestnut trees, and the results  contribute to a better understanding of the physiology of the species in Italian ecosystem conditions.

Would be useful to add a paragraph in the introduction on species representativeness for Europe ecosystem and afterwards to use your case study as Italy.

Maybe a few lines regarding carbon metabolism in the discussion section could support the discussion on carbon storage.

The authors use dentrocronology data relevant to stand. Would have been beneficial to the article to combine such data with  weather and climate information to further model major climate events that shaped those stands in the past. However the article is relevant to its aim.

Author Response

Dear Editor and Referee

We thank the reviewers for their careful reading of the manuscript and constructive remarks. We have taken the comments on board to improve and clarify the manuscript. In particular we have deeply revise the manuscript, changing several paragraph, following reviewers’ suggestions and improving the figures. Further, a native English speaker has revised the paper.

We believe that the manuscript is now more concise and clearer and we hope that the reviewers find it suitable for publication.

The changes are highlighted in the text with Track Changes" function.

Please find below a detailed point-by-point response to all comments.

Looking forward to hearing from you

Giovanna Battipaglia

On behalf of all coauthors

ANSWER TO REVIEWER 1

Comments and Suggestions for Authors

The article is mainly concentrating on dendroisotopic content of chestnut trees, and the results contribute to a better understanding of the physiology of the species in Italian ecosystem conditions.

Would be useful to add a paragraph in the introduction on species representativeness for Europe ecosystem and afterwards to use your case study as Italy.

Thank you for this comment, we added the suggested paragraph

Maybe a few lines regarding carbon metabolism in the discussion section could support the discussion on carbon storage.

Done

The authors use dentrocronology data relevant to stand. Would have been beneficial to the article to combine such data with weather and climate information to further model major climate events that shaped those stands in the past. However the article is relevant to its aim.

The referee is right but unlucky we don’t have stand climate data. We used the best climate information we could find.

Reviewer 2 Report

Dear Authors,

After reading the paper, I think that your manuscript is on an interesting topic but  you need  to valorise your data.  

There are many  issues with the presentation. The Summary section is a narrative and is not clearly understandable. The section of Materials  and Methods is not divide in subsection, making  difficult to assess your work.  Even, the results and discussion session need to be improved significantly. Many editing issues make  the manuscript quite sloppy. In addition, it seems that you compared different stands without replicates. I guess that you can consider the study performed in a single site Monte Amiata, considering even you have only a single climate dataset.

In the title you reported “stand growth  parameters and forest management practices on isotopic content”, but you did not analyze this part, that could be the real core of your manuscript.

Actually I cannot suggest to publish your manuscript, it needs to be really improved.

Below you can find some specific suggestion:

The title is not coherent with the manuscript contents

Line 44 Please, add a reference

Line 55 Please, add a brief description of factors affecting 13C, as you made for 18O

Section 2 Divide in subsection

Fig.2 is not a good quality

Line 74 Is it possible to add information on soil?

Line 94-97 You used the meteorological station of Piancastagnaio, and the data you reported are related to a lower situation, which I guess to be totally different from the study sites. However, I agree to use these data for analysis.

Line 117 Complete the equation

Line 145 Check the references

Line 154 -166 Check the equations format

Line 166 Please, consider the articles Seibt et al. 2008, Gessler et al. 2014, Frank et al. 2015 for more precise definition of WUEi

Line 186 – 190 Please, clarify. If you consider productivity you have to refer to a period.

Line 191-195 If you have only a sampling plot per site, how did you calculate variability to compare stands? If I well understand you want to compare different stands, but the stumps are the statistic unit.

Table 2 Please, reformat. it is very hard to understand. Is it possible to add the results in table1?

Dendrochronology and dendroclimatology, it is very difficult to make dendro using only 14 years.

Fig.4 Please, modify the figure. Residual seems to be the title of x-axys.

Line 241-243 Please, justify your sentence. All the stands show an increasing trend. I don't see any decrease in WUEi of D sites.

Line 247 Do you mean ANCOVA?

Fig.7 Please, verify the stars position.

References 29 and 40 are the same paper

In the discussion section (and maybe in the relationships between isotope and climate), I think that it is not necessary to divide in sites because you are describing the response of trees growing at same elevation, with the same climatic condition and with the same management history. I think in this way, they will be more fluent and describe the physiology of the chestnut in the site.    

Author Response

Dear Editor and Referee

We thank the reviewers for their careful reading of the manuscript and constructive remarks. We have taken the comments on board to improve and clarify the manuscript. In particular we have deeply revise the manuscript, changing several paragraph, following reviewers’ suggestions and improving the figures. Further, a native English speaker has revised the paper.

We believe that the manuscript is now more concise and clearer and we hope that the reviewers find it suitable for publication.

The changes are highlighted in the text with Track Changes" function.

Please find below a detailed point-by-point response to all comments.

Looking forward to hearing from you

Giovanna Battipaglia

On behalf of all coauthors

Comments and Suggestions for Authors

Dear Authors,

After reading the paper, I think that your manuscript is on an interesting topic but  you need  to valorise your data.  

There are many  issues with the presentation.

1.The Summary section is a narrative and is not clearly understandable.

We modified it and we hope that in this current version everything is more understable.

2.The section of Materials  and Methods is not divide in subsection, making  difficult to assess your work.  Even, the results and discussion session need to be improved significantly. Many editing issues make  the manuscript quite sloppy. In addition, it seems that you compared different stands without replicates. I guess that you can consider the study performed in a single site Monte Amiata, considering even you have only a single climate dataset.

We followed review suggestion and the materials and methods section has been divided in subsection. The results and discussion part have been improved. Further, the aim of the study is to compare different stands of chestnut trees, 15 trees for dendrochronological analysis and 5 trees for isotopes, as written in the methods, are considered replicates. All the trees are growing on Monte Amiata but each of them present different structure and management. Following referee suggestion we modified the text, substituting the word “sites” with “stands” since we are comparing different stands in one site “monte Amiata”. The fact that all the stands are in the same site gives us the possibility to use a single climate dataset and to make comparison according to the stand characteristics.

In the title you reported “stand growth  parameters and forest management practices on isotopic content”, but you did not analyze this part, that could be the real core of your manuscript.

You are in right we changed a little the title and we added more comments to exploit forest management aspect

Actually I cannot suggest to publish your manuscript, it needs to be really improved.

We worked hard in this new version of the paper and we hope we manage to satisfy all the referees’ concerns.

Below you can find some specific suggestion:

The title is not coherent with the manuscript contents

We changed it

Line 44 Please, add a reference

Done

Line 55 Please, add a brief description of factors affecting 13C, as you made for 18O

Done

Section 2 Divide in subsection

Done

Fig.2 is not a good quality

We modified the figure

Line 74 Is it possible to add information on soil?

Done

Line 94-97 You used the meteorological station of Piancastagnaio, and the data you reported are related to a lower situation, which I guess to be totally different from the study sites. However, I agree to use these data for analysis.

We used the only data available in that area. It is very rare to have meteo stations at high elevation.

Line 117 Complete the equation

To clarify the equation, we modified the whole paragraph. We hope that now the WUEi calculation is clear.

Line 145 Check the references

Done

Line 154 -166 Check the equations format

Done

Line 166 Please, consider the articles Seibt et al. 2008, Gessler et al. 2014, Frank et al. 2015 for more precise definition of WUEi

Done.

Line 186 – 190 Please, clarify. If you consider productivity you have to refer to a period.

Done.

Line 191-195 If you have only a sampling plot per site, how did you calculate variability to compare stands? If I well understand you want to compare different stands, but the stumps are the statistic unit.

In each stand we have several trees that are out statistic unit. Thus the variability refers to the variability calculate among trees sampled in the same stands

1

Table 2 Please, reformat. it is very hard to understand. Is it possible to add the results in table1?

Table 1 is already very complex and we believe that adding results would make the discussion even more difficult to follow. We would prefer to leave in table 1 only the reported information

Dendrochronology and dendroclimatology, it is very difficult to make dendro using only 14 years.

The referee is right, infact we did not apply some statistical procedures such as boot-strap often used in dendroclimatology with long series. That is the reason why we apply a different approach combining classic dendro with isotopes and climate correlations.

Fig.4 Please, modify the figure. Residual seems to be the title of x-axys.

Done.

Line 241-243 Please, justify your sentence. All the stands show an increasing trend. I don't see any decrease in WUEi of D sites.

Done.

Line 247 Do you mean ANCOVA?

No ANOVA

Fig.7 Please, verify the stars position

Done.

.

References 29 and 40 are the same paper

Sorry for this mistake. We modified it accordingly.

In the discussion section (and maybe in the relationships between isotope and climate), I think that it is not necessary to divide in sites because you are describing the response of trees growing at same elevation, with the same climatic condition and with the same management history. I think in this way, they will be more fluent and describe the physiology of the chestnut in the site.    

Indeed, all the discussion is based on the possible influence of stand characteristics and on how those characteristics can influence the responses to climate. If we merge the sites we could lose an important part of information. We will take into account your suggestion when we will consider the differences among the provenances, hopefully in the next future.